# Clinical features of and severity risk factors for COVID-19 in adults during the predominance of SARS-CoV-2 XBB variants in Okinawa, Japan

Shuhei Ideguchi[1], Kazuya Miyagi[1]*, Wakaki Kami[2], Daisuke Tasato[3], Futoshi Higa[4], Noriyuki Maeshiro[5], Shota Nagamine[5], Hideta Nakamura[1], Takeshi Kinjo[1], Masashi Nakamatsu[6], Shusaku Haranaga[7], Akihiro Tokushige[8], Shinichiro Ueda[8], Jiro Fujita[2], Kazuko Yamamoto[1]

1 Division of Infectious, Respiratory, and Digestive Medicine, First Department of Internal Medicine, University of the Ryukyus Graduate School of Medicine, Okinawa, Japan, 2 Department of Respiratory Medicine, Ohama Dai-ichi Hospital, Okinawa, Japan, 3 Department of Respiratory and Infectious Diseases, Okinawa North Medical Association Hospital, Okinawa, Japan, 4 Department of Respiratory Medicine, NHO Okinawa Hospital, Okinawa, Japan, 5 Research Center for Infectious Diseases, Okinawa Prefectural Institute of Health and Environment, Okinawa, Japan, 6 Department of Infection Control, University of the Ryukyus Hospital, Okinawa, Japan, 7 University of the Ryukyus Comprehensive Health Professions Education Center, University Hospital, Okinawa, Japan, 8 Department of Clinical Pharmacology and Therapeutics, Graduate School of Medicine, University of the Ryukyus, Okinawa, Japan

* myagik@med.u-ryukyu.ac.jp

**Data Availability Statement:** All relevant data are within the article and its supporting information files.

## Abstract

### Background and objective

Since 2023, COVID-19 induced by SARS-CoV-2 XBB variants have been a global epidemic. The XBB variant-induced epidemic was largest in the Okinawa Prefecture among areas in Japan, and healthcare institutions have been burdened by increased COVID-19 hospitalizations. This study aimed to evaluate the clinical features of XBB variant-induced COVID-19 and risk factors for severe COVID-19.

### Methods

This retrospective study included adult patients hospitalized for COVID-19 between May and July 2023 at four tertiary medical institutions in Okinawa, Japan. Patients with bacterial infection-related complications were excluded. According to oxygen supplementation and intensive care unit admission, patients were divided into two groups, mild and severe. Patient backgrounds, symptoms, and outcomes were compared between both groups, and the risk factors for severe COVID-19 were analyzed using a multivariate logistic regression model.

### Results

In total of 367 patients included, the median age was 75 years, with 18.5% classified into the severe group. The all-cause mortality rate was 4.9%. Patients in the severe group were more older, had more underlying diseases, and had a higher mortality rate (13.2%) than those in the mild group (3.0%). Multivariate logistic regression analysis showed that

**Funding:** S.I. and K.Y. are funded by a research grant from the Advanced Medical Research Center, Faculty of Medicine, University of the Ryukyus, Japan (https://amrc.skr.u-ryukyu.ac.jp/aboutus/). The funder had no role in study design, data collection and analysis, decision to publish, or preparation of the manuscript.

**Competing interests:** The authors have declared that no competing interests exist.

**Abbreviations:** SARS-CoV-2, severe acute respiratory syndrome coronavirus 2; COVID-19, coronavirus disease 2019; PCR, polymerase chain reaction; BMI, body mass index; SpO2, oxygen saturation; OR, odds ratio; CI, confidence interval; anti-N, anti-nucleocapsid..

diabetes mellitus was an independent risk factor for severe COVID-19 (95% confidence interval [CI], 1.002–3.772), whereas bivalent omicron booster vaccination was an independent factor for less severe COVID-19 (95% CI, 0.203–0.862).

## Conclusion

This study implies that assessing risk factors in older adults is particularly important in the era of omicron variants.

## Introduction

COVID-19 epidemic has been induced by new omicron subvariants of SARS-CoV-2 worldwide. Since 2023, the XBB variant, the recombination of two BA.2 lineages (BJ.1 and BA.2.75), has become the most prevalent variant [1–3].

XBB variants have virological characteristics, such as stronger binding to human angiotensin-converting enzyme 2, higher evading ability from infection-derived antibodies than previous omicron variants, and resistance to clinically authorized therapeutic antibodies other than sotrovimab [4–8]. Few studies have reported the real-world clinical features of XBB variant-induced COVID-19 and some studies have shown that patients infected with XBB variants were more likely to have a past history of SARS-CoV-2 infection than those infected with non-XBB variant, whereas no differences were observed in the severity or hospitalization rate [9, 10]. Older age, unvaccinated status, immunosuppression, and underlying heart, kidney, and lung diseases have been associated with the risk of hospitalization for XBB variant-induced COVID-19 [11]. However, since the beginning of the omicron strain-induced epidemic, risk factors for severe COVID-19 have not been re-evaluated, which is crucial to reflecting their perceived importance in preventing severe COVID-19, and no real-world data on elderly patients with XBB variant-induced COVID-19 is available to date.

Okinawa, the island located in the southernmost prefecture of Japan, where 10 million tourists visit annually, experienced frequent episodes of the COVID-19 epidemic before mainland Japan [12]. The infectious disease law in Japan changed from total surveillance to fixed-point surveillance of COVID-19 on May 8, 2023, making it difficult to evaluate the clinical characteristics of COVID-19. The ninth wave of the pandemic occurred in Okinawa before the mainland between May 2023 and July 2023 [13]. The number of hospitalized patients increased during the ninth wave compared with the previous epidemic wave, and hospital beds in Okinawa were tightened [14]. Genomic surveillance conducted by the Institute of Public Health and Environment in Okinawa between May and July 2023 revealed that approximately 80% of SARS-CoV-2 strains were XBB variants, including XBB.1.5 (5.7–25.9%), XBB.1.16 (9.4–43.8%), XBB.1.9.1 (9.4–31.1%), XBB.1.9.2 (1.2–9.4%), and XBB.2.3 (3.5–28.2%) [15].

This study aimed to evaluate the clinical characteristics of patients hospitalized for XBB variant-induced COVID-19 in Okinawa and assessed the risk factors for severe COVID-19.

## Methods

### Patients and study design

This retrospective study was conducted at four tertiary hospitals (University of the Ryukyus Hospital, Ohama Daiichi Hospital, NHO Okinawa Hospital, and Okinawa North Medical Association Hospital) in Okinawa between May and July 2023. Inclusion criteria were age ≥18

years, and patients required hospitalization due to COVID-19. The diagnostic criteria for COVID-19 were as follows: SARS-CoV-2 infection confirmed using antigen or polymerase chain reaction (PCR) testing, and the presence of acute COVID-19 symptoms, included fever, fatigue, dyspnea, cough, sputum, nasal discharge, sore throat, headache, diarrhea, and a loss of smell or taste sensation. The exclusion criteria were bacterial co-infection, including bacterial pneumonia and bacteremia, a history of tracheotomy, and ventilator use. Complications of bacterial pneumonia were defined as the presence of new lung infiltrates on chest radiography, neutrophils on sputum smears with a positive culture, and treatment with antimicrobial agents.

## Whole-genome sequencing of SARS-CoV-2

For patients admitted to the University of the Ryukyus Hospital, reverse transcription-quantitative PCR (RT-qPCR) testing for SARS-CoV-2 was performed using frozen nasopharyngeal specimens stored at -80˚C. Positive RNA samples (Ct value< 35.0) were subjected to whole-genome sequencing. Whole-genome sequencing of SARS-CoV-2 was performed through a combination of multiplex PCR and next-generation sequencing (NGS), referring to ARTIC Network's modified protocol [16]. The PCR products in pools 1 and 2 from the same clinical sample were pooled, purified, and subjected to Illumina library construction using a QIAseq FX DNA library kit (QIAGEN, Hilden, Germany). The iSeq 100 platform (Illumina, San Diego, CA) was used for sequencing the indexed libraries. The NGS reads were mapped to the SARS-CoV-2 Wuhan-Hu-1 reference genome sequence (29.9-kb single-stranded RNA [ssRNA] [GenBank accession no. MN908947.3]). The specimen-specific SARS-CoV-2 genome sequence was obtained by complete mapping to the reference sequence. The SNV sites and marked heterogeneity were extracted by read mapping at a $\geqq$10 x depth from the region spanning nucleotides (nt) 55 to 29836 of the Wuhan-Hu-1 genome sequence. We used CLC Genomics Server ver.23.0.5 and CLC Genomics Workbench ver.23.0.5 (https://digitalinsights.qiagen.com/) in this analysis involving lineage classification.

## Data collection

Patient characteristics, COVID-19 severity, and outcomes were extracted from the hospitals' electronic medical records. The following information was collected: age, sex, body mass index (BMI), smoking status, hospital-acquired infection status, SARS-CoV-2 vaccination status, underlying disease, medication use, medical history before admission, symptoms, clinical variables, and outcomes. Hospital-acquired infection was defined as new onset of COVID-19 during hospitalization for other diseases. The SARS-CoV-2 vaccination status included all vaccine types: BNT162b2, mRNA-1273, AZD1222, NVX-CoV2373, and a bivalent omicron booster vaccine (ancestral and BA.4/5). Based on the severity of COVID-19, patients were divided into two groups: "severe group" requiring oxygen supplementation or intensive care unit admission and "mild group" requiring none of the above. Outcomes were evaluated based on deaths during hospitalization.

## Statistical analysis

All statistical analyses were performed using IBM® SPSS® version 28 (IBM Corp., Armonk, NY, USA). Between-group differences in the distribution of values were evaluated using the Mann–Whitney U test, and the null hypothesis of independence between two binomial variables was tested using a two-sided Fisher's exact test. Multivariate analysis of previously reported risk factors and current factors identified in the present study was performed, indicating a *p*-value <0.1 for severe disease using the Logistic Regression model.

## Human/animal ethics approval declaration

This study was approved by the ethics committee of the University of the Ryukyus for Medical and Health Research Involving Human Subjects (Approval number "23–2149-00-00-00"). Informed consent for patients was obtained in the form of opt-out on the hospital website (http://www.hosp.u-ryukyu.ac.jp/information/optout.html).

## Results

### Patient characteristics

Of the 458 patients hospitalized for COVID-19 during the study period, 367 were enrolled (Fig 1). The number of enrolled patients in each hospital was as follows: University of the Ryukyus Hospital (n = 101), Ohama Daiichi Hospital (n = 82), NHO Okinawa Hospital (n = 54), and Okinawa North Medical Association Hospital (n = 130). Of 91 patients excluded from the study, 75 patients had bacterial pneumonia, and 13 had bacteremia without pneumonia. Patient characteristics are presented in Table 1. Overall, the median age was 75 years, and half of the patients were male. The median BMI was 22.3 kg/m$^2$, and none of the patients were extremely obese. Only 9% of the patients (n = 32) were current smokers, and hospital-acquired infections accounted for 38% of cases (n = 140). Of 359 patients, 36 (10%) had a history of SARS-CoV-2 infection. SARS-CoV-2 vaccination status was evaluated in 344 patients; 20% (n = 68) were unvaccinated, and 27% (n = 94) had received bivalent booster vaccination. The most common underlying disease was hypertension (52.3%), followed by cardiovascular disease (26.2%), malignancy (25.9%), dyslipidemia (23.7%), chronic lung disease (22.3%), and diabetes mellitus (20.7%). Systemic corticosteroids were the most commonly used immunosuppressants (n = 39 [10%]), followed by multiple immunosuppressants (n = 14 [4%]). Notably, 17% (n = 64) of the patients visited a medical clinic and were diagnosed with COVID-19 before admission; however, only 28% (n = 18) were prescribed oral anti-SARS-CoV-2 medications. The median time from the onset of the disease to the initiation of COVID-19 treatment was one day. Comparing the mild and severe groups, patients in the severe group were older and had a lower rate of in-hospital onset than those in the mild group. The bivalent omicron booster vaccination rates tended to be lower in the severe group. Regarding underlying

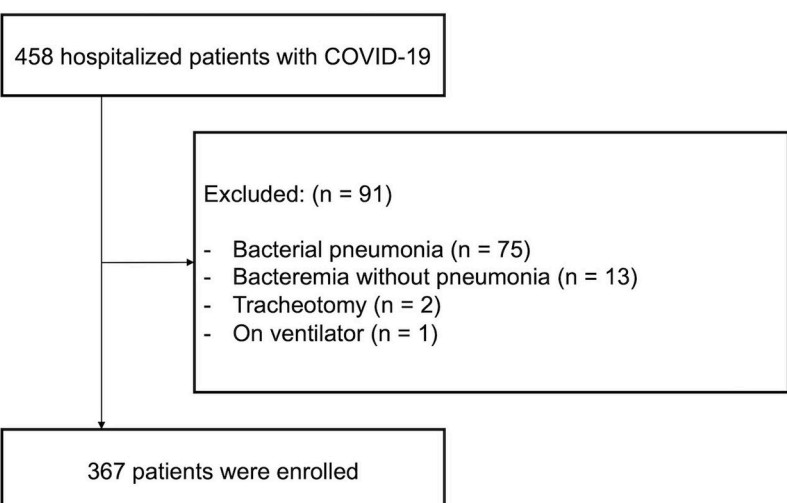

**Fig 1. Flow chart of the selection process of the eligible patients.**

**Table 1. Patient characteristics.**

| | Total (n = 367) | | Mild group (n = 299) | | Severe group (n = 68) | | p value |
|---|---|---|---|---|---|---|---|
| Age | 75 | (65–86) | 74 | (64–85) | 84 | (70–91) | <0.001 |
| Male/Female | 185/182 | (50/50) | 148/151 | (49/51) | 37/31 | (54/46) | 0.503 |
| Body mass index | 22.3 | (19.5–25.7) | 22.4 | (19.6–25.4) | 22.0 | (19.3–27.0) | 0.905 |
| Current smoking | 32 | (9) | 26 | (9) | 6 | (9) | 1.000 |
| Hospital-acquired infection | 140 | (38) | 125 | (42) | 15 | (22) | 0.002 |
| History of SARS-CoV-2 infection[†] | 36 | (10) | 28 | (10) | 8 | (12) | 0.502 |
| SARS-CoV-2 vaccination status[§] | | | | | | | |
| Number of vaccinations | 3 | (2–5) | 4 | (2–5) | 3 | (2–4) | 0.217 |
| Unvaccinated | 68 | (20) | 56 | (20) | 12 | (19) | 0.864 |
| Bivalent booster vaccination | 94 | (27) | 82 | (29) | 12 | (19) | 0.089 |
| Underlying disease[♭] | | | | | | | |
| Hypertension | 192 | (52) | 147 | (49) | 45 | (66) | 0.015 |
| Cardiovascular disease | 96 | (26) | 69 | (23) | 27 | (40) | 0.009 |
| Malignancy | 95 | (26) | 86 | (29) | 9 | (13) | 0.009 |
| Dyslipidemia | 87 | (24) | 66 | (22) | 21 | (31) | 0.154 |
| Chronic lung disease | 82 | (22) | 61 | (20) | 21 | (31) | 0.075 |
| Diabetes mellitus | 76 | (21) | 53 | (18) | 23 | (34) | 0.005 |
| Cerebrovascular disease | 72 | (20) | 57 | (19) | 15 | (22) | 0.612 |
| Chronic kidney disease | 69 | (19) | 54 | (18) | 15 | (22) | 0.492 |
| Autoimmune disease | 30 | (8) | 26 | (9) | 4 | (6) | 0.624 |
| Chronic liver disease | 20 | (5) | 18 | (6) | 2 | (3) | 0.552 |
| Transplantation | 3 | (1) | 2 | (1) | 1 | (2) | 0.460 |
| Medication[··] | | | | | | | |
| Systemic corticosteroid | 39 | (11) | 32 | (11) | 7 | (10) | 1.000 |
| Calcineurin inhibitor | 8 | (2) | 5 | (2) | 3 | (4) | 0.170 |
| Multiple immunosuppressant use | 14 | (4) | 11 | (4) | 3 | (4) | 0.729 |
| Clinic visits prior to admission | 64 | (17) | 48 | (16) | 16 | (24) | 0.157 |
| Oral anti-SARS-CoV-2 agent prior to admission | 18 | (5) | 11 | (4) | 7 | (10) | 0.054 |
| Molnupiravir | 9 | (3) | 4 | (2) | 5 | (9) | 0.012 |
| Nirmatrelvir/ritonavir | 4 | (1) | 3 | (1) | 1 | (2) | 0.550 |
| Ensitrelvir | 4 | (1) | 3 | (1) | 1 | (2) | 0.550 |
| From onset to initiation of anti-SARS-CoV-2 agents, day | 1 | (0–2) | 1 | (0–1) | 1 | (0–2) | 0.226 |

Data are presented as frequency (%) or median (interquartile range).

[†]History of SARS-CoV-2 infection was evaluated in 359 patients

[§]SARS-CoV-2 vaccination status was evaluated in 344 patients.

[♭]One case each of pregnancy and HIV.

[··]One case each of biologics and mycophenolate mofetil use.

diseases, the severe group had a higher incidence of hypertension, cardiovascular diseases, and diabetes mellitus and a lower incidence of malignancies than the mild group. Pre-admission antiviral therapy was more common in the severe group, particularly a molnupiravir prescription.

## Distribution of SARS-CoV-2 variants

Only patients hospitalized at the University of the Ryukyus Hospital in this study had stored nasopharyngeal samples. Of 101 patients admitted to this hospital, five patients had no stored

specimens, and samples of eight patients could not be analyzed due to variants being below the limit of detection; finally, the stored specimens of 86 patients were genetically analyzed for identifying SARS-CoV-2 variants. SARS-CoV-2 omicron sub-lineages were identified as XBB (n = 56, 65.1%), BA.2 (n = 20, 23.3%), BA.2.75 (n = 8, 9.3%), and BA.5 (n = 2, 2.3%) by GISAID. Among 56 XBB variants, according to Nextclade_pango lineage classification, XBB.1.5.1 (n = 23, 26.7%) and XBB.1.16 (n = 12, 14.0%) were identified as predominant sub-lineages in addition to other 18 kinds of variants (total n = 21) (S1 Data).

## Symptoms, clinical variables, and outcomes

Data on symptoms, COVID-19 severity, the ratio of percutaneous $SpO_2$ to fractional inspired oxygen, and deaths during hospitalization are presented in Table 2. Overall, the most common symptom was fever, followed by cough, fatigue, and sputum production. Regarding COVID-19 severity, >50% of the patients had mild (62.9%, n = 231) severity, whereas 18% (n = 66) had moderate II severity requiring oxygen administration. Only a few patients (0.5%, n = 2) required ventilator use. The mortality rate during hospitalization was 4.9% (n = 18); 2.2% (n = 8) owing to COVID-19 and 2.7% (n = 10) owing to other reasons. A higher prevalence of dyspnea and sputum production and a lower prevalence of headaches was observed in the severe group than in the mild group. The mortality rate was significantly higher in the severe group than in the mild group, regardless of the cause of death (13.2% vs. 3.0%).

**Table 2. Symptoms, severity, and outcomes.**

| | Total (n = 367) | | Mild group (n = 299) | | Severe group (n = 68) | | p value |
|---|---|---|---|---|---|---|---|
| Fever | 310 | (85) | 248 | (83) | 62 | (91) | 0.098 |
| Cough | 247 | (67) | 199 | (67) | 48 | (71) | 0.569 |
| Fatigue | 198 | (54) | 165 | (55) | 33 | (49) | 0.347 |
| Sputum | 149 | (41) | 113 | (38) | 36 | (53) | 0.028 |
| Sore throat | 112 | (31) | 95 | (32) | 17 | (25) | 0.309 |
| Dyspnea | 78 | (21) | 50 | (17) | 28 | (41) | <0.001 |
| Nasal discharge | 63 | (17) | 54 | (18) | 9 | (13) | 0.379 |
| Diarrhea | 50 | (14) | 45 | (15) | 5 | (7) | 0.117 |
| Headache | 49 | (13) | 45 | (15) | 4 | (6) | 0.048 |
| New loss of sense of taste | 14 | (4) | 10 | (3) | 4 | (6) | 0.303 |
| New loss of sense of smell | 3 | (1) | 1 | (0.3) | 2 | (3) | 0.089 |
| Severity of COVID-19[∂] | | | | | | | |
| Mild | 231 | (63) | 231 | (77) | 0 | (0) | |
| Moderate I | 68 | (19) | 68 | (23) | 0 | (0) | |
| Moderate II | 66 | (18) | 0 | (0) | 66 | (97) | |
| Severe | 2 | (1) | 0 | (0) | 2 | (3) | |
| $SpO_2/FiO_2$ ratio | 452 | (433–462) | 457 | (448–462) | 386 | (335–410) | <0.001 |
| Deaths during hospitalization | 18 | (5) | 9 | (3) | 9 | (13) | 0.002 |
| Death due to COVID-19 | 8 | (2) | 3 | (1) | 5 | (7) | |
| Death due to non-COVID-19 | 10 | (3) | 5 | (2) | 5 | (7) | |

Data are presented as frequency (%) or median (interquartile range).

[∂] COVID-19 severity was defined according to the Japanese Ministry of Health, Labour and Welfare's COVID-19 Medical Care Guidelines (version 9.0).

$SpO_2/FiO_2$, ratio of percutaneous oxygen saturation to fractional inspired oxygen.

### Analysis of risk factors for COVID-19 severity

After excluding 23 cases with partially missing values, the remaining 344 cases underwent multivariate logistic regression analysis. Table 3 presents the results of this analysis. Diabetes mellitus was an independent risk factor for severe COVID-19 (odds ratio [OR]: 1.944; 95% confidence interval [CI]: 1.002–3.772; $p$ = 0.049). In contrast, bivalent omicron booster vaccination was an independent factor that reduced the risk of severe COVID-19 (OR, 0.419; 95% CI, 0.203–0.862; $p$ = 0.018).

## Discussion

The present study revealed the association between prior bivalent omicron booster vaccination and lower severity of XBB variant-induced COVID-19 among hospitalized adult patients in Okinawa, Japan. Previous studies have shown that the bivalent omicron booster vaccine enhanced the neutralizing ability and humoral immunity against XBB variants compared with existing vaccines in vitro [17]. In the United States, a test-negative case-control study in a large health system showed that the BNT162b2 BA.4/5 bivalent booster vaccine restored protection against XBB variant-induced COVID-19, especially against hospital admission and critical illness [18]. In contrast, some studies have reported that the bivalent omicron booster vaccine did not reduce the risk of COVID-19 incidence or hospitalization [10, 19]. The present study could not evaluate the vaccine-induced reduction of COVID-19 incidence or hospitalization risk because only hospitalized patients were included; however, we could confirm the bivalent omicron booster vaccine-induced prevention of severe illness after hospitalization during the XBB variant-induced pandemic. In this study, most of the eligible patients were older adults. The age distribution in the studies in other countries showing the efficacy of the bivalent omicron booster vaccine differed from that in Japan, with 15–30% of adults being ≥65 years old [18, 20]. Further studies are needed to determine whether the bivalent omicron booster vaccine also reduces the severity of COVID-19 in terms of hospitalizations in younger adults. The bivalent omicron booster vaccine coverage in the present study was <30%, which requires further education for elderly, particularly those with underlying diseases, on the need for future vaccination against XBB variants because Okinawa Prefecture has a low SARS-CoV-2 vaccination rate compared with other areas of Japan [21].

In the present study, 19% of patients (n = 88) were excluded because of concurrent bacterial infections. The high co-infection rate in the present study compared with that in previous

**Table 3. Multivariate logistic regression analysis of risk factors for COVID-19 severity.**

| Variables | Odds ratio | 95% CI | $p$ value |
|---|---|---|---|
| Age | 1.018 | 0.996–1.041 | 0.112 |
| Male | 1.122 | 0.623–2.019 | 0.701 |
| Hospital acquired | 0.515 | 0.255–1.040 | 0.064 |
| Bivalent booster vaccination | 0.419 | 0.203–0.862 | 0.018 |
| Hypertension | 1.431 | 0.741–2.763 | 0.286 |
| Cardiovascular diseases | 1.582 | 0.837–2.990 | 0.158 |
| Malignancy | 0.525 | 0.229–1.201 | 0.127 |
| Chronic lung disease | 1.297 | 0.664–2.535 | 0.447 |
| Diabetes mellitus | 1.944 | 1.002–3.772 | 0.049 |
| Oral anti-SARS-CoV-2 agent prior to admission | 2.142 | 0.740–6.207 | 0.160 |

CI, confidence interval.

studies could be attributed to the older patient population and the limitations of inpatients in the present study [22]. The present study also had a low proportion of patients with a history of SARS-CoV-2 infection. In May 2023, Ministry of Health, Labour and Welfare conducted a surveillance of anti-nucleocapsid (anti-N) antibody possession using residual serum from blood donations throughout Japan and showed that the anti-N antibody possession rate in Okinawa was 58.9%, indicating that more than half of the people living in Okinawa had a history of COVID-19 [23]. However, the actual proportion of those with a history of COVID-19 in the present study was difficult to assess because anti-N antibody possession rates decrease with increasing age [23, 24]. This suggested that older people had more infection-avoiding behaviors.

Despite the large proportion of elderly patients in the present study, only 28% of patients diagnosed with COVID-19 in the clinic before admission were prescribed anti-SARS-CoV-2 agents. This indicates that awareness of the treatment of patients with mild COVID-19 who had risk factors for severe disease was not expanded in clinics or that clinicians underestimated the COVID-19 caused by omicron variants as being less severe. A previous study showed that in elderly patients aged ≥60 years, nirmatrelvir/ritonavir reduced the risk of hospitalization due to COVID-19 caused by omicron strains, including XBB variants [25]. Educating clinicians and standardizing early treatment interventions for COVID-19 is an urgent issue. The high number of molnupiravir prescriptions could be due to unnecessary volume adjustments based on renal function and/or less concerning drug interactions.

The present study included a large proportion of patients with underlying malignancies (25.9%). A previous study showed that XBB-induced COVID-19 was significantly more common in patients with malignancy than in those with BA.5.2, suggesting that there could be an association between XBB variants and malignancy [26]. Regarding symptoms, loss of smell or taste sense was rarely observed. Fever, fatigue, cough, sputum, and sore throat were similar symptoms to those of other upper respiratory viral infections in the mild group. Dyspnea was more frequent in the severe group; therefore, diagnosing and treating COVID-19 as early as possible without overlooking it is important. No cases of simultaneous influenza virus infections were observed in the present study.

Severe COVID-19 caused by XBB variants is uncommon, similar to that caused by previous omicron variants [9, 10, 25, 27]; however, the mortality rate in the present study was as high as 4.9%, possibly due to the fact that the study included only hospitalized patients, and many of the patients were older adults with underlying diseases.

Multivariate logistic regression analysis revealed diabetes mellitus as an independent risk factor for severe COVID-19. Previous studies have reported diabetes as one of the strongest risk factors for severe COVID-19, and the presence of hyperglycemia without diabetes has also been reported as a risk factor for COVID-19 [28–30]. However, since the omicron strain epidemic, few reports have re-evaluated the risk factors for severe COVID-19 and their assessment. In a US-based observational study on COVID-19 caused by omicron strains, including XBB, diabetes was not a risk factor for hospitalization [10]. The present study relays implications for the assessment of risk factors for severe COVID-19 among elderly since the omicron variants became endemic. These were not statistically independent risk factors for severe COVID-19; however, hypertension, cardiovascular disease, dyslipidemia, chronic lung disease, cerebrovascular disease, and chronic kidney disease showed higher comorbidity rates in the severe group, similar to the results for COVID-19 caused by previously reported SARS-CoV-2 variants [11, 31]. The higher proportion of patients with malignancies in the mild group could be due to higher hospital-acquired infection rates (65%), and patients with malignancy were more likely to visit the hospital frequently, even in community-acquired cases, which could have led to early diagnosis and treatment of COVID-19.

The present study had several limitations. First, the method used for confirming SARS-CoV-2 infection varied, and antigen testing or PCR was used depending on the facility. However, antigen testing is known to be as specific as PCR for Omicron variants and is used at many facilities in Japan for the diagnosis of COVID-19 [32]. Second, this study included only hospitalized patients and did not evaluate all the clinical characteristics of XBB variant-induced COVID-19. Third, the actual proportion of XBB variant-induced COVID-19 cases was unknown because genome analysis was not performed in all cases. However, the Institute of Public Health and Environment performed genome analysis of specimens from patients diagnosed with COVID-19 in Okinawa once a week, and consistent with the findings of our analysis, XBB variant-induced COVID-19 continuously accounted for the most cases during the ninth wave of the pandemic [15]. Furthermore, differences among XBB subvariants could have affected the result of the present study because the risk of severe outcomes was reported to be higher in cases of XBB.1.16 infection than in those of XBB.1.5 or XBB.1.9 infection, especially in older adults [33]. Fourth, we could not evaluate the time between the last SARS-CoV-2 vaccination and onset of COVID-19. The SARS-CoV-2 vaccine attenuates the effects of infection, hospitalization risk, and COVID-19 severity depending on the time since vaccination, which may have influenced the results of this study [34–36]. Fifth, the exclusion of obvious bacterial co-infections in this study could have affected the results of the analysis. However, our objective was to evaluate the risk factors for severe COVID-19 itself.

## Conclusion

This study revealed the characteristics of hospitalized adult patients with XBB variant-induced COVID-19. Education on outpatient medical treatment at clinics and awareness of vaccination of older adults with underlying diseases are critical issues that should be addressed to prepare for the next pandemic of COVID-19. Furthermore, this study implies that re-evaluating risk factors in the elderly is particularly important in the era of omicron variants.

## Supporting information

**S1 Data. Whole-genome sequencing.**
(XLSX)

**S2 Data. Relevant data.**
(XLSX)

## Acknowledgments

We would like to express our special thanks to Makoto Ohnishi MD. PhD., Okinawa Prefectural Institute of Health and Environment Research Center for Infectious Diseases, for his support in whole-genome sequencing. We also thank Shigeki Chinen MD., Haruka Zukeyama MD., Kenta Kuniyoshi MD., Michika Setoguchi MD., Kazutaka Yamaniha MD., Tomo Kiyuna MD., Nanae Ikemiyagi MD., Shoshin Yamazato MD., Tomoko Yamashiro MD., Daijiro Nabeya MD. PhD., Makoto Furugen MD. PhD., Rio Sugino, Chiho Nozaki, Sae Furuhashi, all of whom in University of the Ryukyus for their valuable support with clinical data input. NGS data analysis was performed using the PathoGenS: Pathogen Genomic data collection System with CLC Genomics Server and CLC Genomics Workbench. The PathoGenS is a system for pathogen genomic surveillance in the National Epidemiological Surveillance of Infectious Diseases (NESID) Program, and is supported by Grant-in-Aid for Research on Emerging and Reemerging Infectious Diseases and Immunization from the Ministry of Health, Labour and Welfare, Japan [grant numbers JPMH21HA2003, JPMH24HA2005].

## Author Contributions

**Conceptualization:** Shuhei Ideguchi, Kazuya Miyagi, Kazuko Yamamoto.

**Data curation:** Shuhei Ideguchi, Kazuya Miyagi, Wakaki Kami, Daisuke Tasato, Futoshi Higa, Hideta Nakamura, Takeshi Kinjo, Jiro Fujita, Kazuko Yamamoto.

**Formal analysis:** Shuhei Ideguchi, Akihiro Tokushige, Shinichiro Ueda.

**Funding acquisition:** Shuhei Ideguchi, Kazuko Yamamoto.

**Investigation:** Shuhei Ideguchi, Kazuya Miyagi, Wakaki Kami, Daisuke Tasato, Futoshi Higa, Noriyuki Maeshiro, Shota Nagamine, Hideta Nakamura, Masashi Nakamatsu, Jiro Fujita, Kazuko Yamamoto.

**Methodology:** Shuhei Ideguchi, Kazuya Miyagi, Noriyuki Maeshiro, Shota Nagamine, Kazuko Yamamoto.

**Project administration:** Kazuya Miyagi.

**Software:** Shuhei Ideguchi, Noriyuki Maeshiro, Shota Nagamine.

**Supervision:** Shuhei Ideguchi, Kazuya Miyagi, Kazuko Yamamoto.

**Validation:** Shuhei Ideguchi, Kazuya Miyagi, Wakaki Kami, Daisuke Tasato, Futoshi Higa, Noriyuki Maeshiro, Shota Nagamine, Hideta Nakamura, Takeshi Kinjo, Masashi Nakamatsu, Shusaku Haranaga, Jiro Fujita, Kazuko Yamamoto.

**Visualization:** Shuhei Ideguchi.

**Writing – original draft:** Shuhei Ideguchi.

**Writing – review & editing:** Kazuya Miyagi, Takeshi Kinjo, Masashi Nakamatsu, Shusaku Haranaga, Akihiro Tokushige, Shinichiro Ueda, Kazuko Yamamoto.

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
