## [Decision Letter · Decision Letter 0]

20 May 2024

PONE-D-24-16162Clinical features of and severity risk factors for COVID-19 in adults during the epidemic wave induced by SARS-CoV-2 XBB variants in Okinawa, JapanPLOS ONE

Dear Dr. Miyagi,

Thank you for submitting your manuscript to PLOS ONE. After careful consideration, we feel that it has merit but does not fully meet PLOS ONE’s publication criteria as it currently stands. Therefore, we invite you to submit a revised version of the manuscript that addresses the points raised during the review process.

We look forward to receiving your revised manuscript.

Kind regards,

Liling Chaw

Academic Editor

PLOS ONE

Journal Requirements:

"S.I. and K.Y. are funded by a research grant of Advanced Medical Research Center, Faculty of Medicine, University of the Ryukyus, Japan.

There is no grant number.

URL: https://amrc.skr.u-ryukyu.ac.jp/aboutus/

The sponsor played no role in this study."

Additional Editor Comments:

Dear authors,

Please address the reviewer's comments, particularly on the definition of XBB variant and justifying the exclusion criteria used.

Reviewers' comments:

Reviewer's Responses to Questions

**Comments to the Author**

1. Is the manuscript technically sound, and do the data support the conclusions?

Reviewer #1: Yes

Reviewer #2: Partly

2. Has the statistical analysis been performed appropriately and rigorously? 

Reviewer #1: I Don't Know

Reviewer #2: Yes

3. Have the authors made all data underlying the findings in their manuscript fully available?

Reviewer #1: Yes

Reviewer #2: Yes

4. Is the manuscript presented in an intelligible fashion and written in standard English?

Reviewer #1: Yes

Reviewer #2: Yes

5. Review Comments to the Author

Reviewer #1: It is an interesting manuscript. The authors described the characteristics of patients, symptoms and signs, as well as the risk factors of COVID-19 patients in Okinawa. Although the titles mention about SARS-CoV-2 XBB variants, the paper lack of information about the methods for the XBB variants identification in the patients enrolled in this study. Learning that XBB variant correspond to 80% of the genomic surveillance does not reflect directly to the proportion among the patients enrolled in this study. I advise not to include XBB variants in the title, since it will be misleading to the readers.

Moreover, there are several suggestions to improve this manuscript:

1. Lines 64: This phrase cannot clearly understandable “….XBB/XBB1.5 had more prior infections than…. “

2. Fig 1. Should be explained in the result, but not in the introduction. The methods to collect the data need to be describe clearly in the methods. Otherwise, cite from the original publication.

3. Line 97: There is some disagreement that antigen testing can be use as confirmatory diagnosis for COVID-19. Authors need to address this in discussion with solid references.

4. Lines 99-100: Please describe the diagnosis criteria for bacterial pneumoniae. Is it supported with culture from blood or respiratory specimen? Or other methods such as NAAT?

5. Line 46: How authors defined hospital acquired infection for COVID-19? Does the author want to explain that 38% of the subjects contracted with COVID-19 while they were stayed in the hospital because of suffering from other diseases? Or, it means that 38% of the patients have other HAIs (not the COVID-19)

6. Table 1. Why only male is presented?

7. This study only recruited elderly (64 – 91) patients. Though it may reflect the actual hospitalization cases in Okinawa, the author should address these issues. Authors should refer to other countries reports which may have different segment of patients ages. This is important to give comprehensive insight into the characteristics, symptoms and sign, as well as risk factors for the XBB variant.

Reviewer #2: The study examined clinical features of COVID-19 caused by the XBB variant of SARS-CoV-2 and explored risk factors for severe disease among hospitalized patients.

The study was done in 4 tertiary care hospitals in Japan. Patients with complications related to bacterial infections were excluded.

Exclusion criteria appear to be complications that one would expect to occur in severe disease, so it is not clear why those who had these outcomes were excluded. In a cohort study, exclusions are made based on factors knowable before day 1 for each patient, not on factors that occur after day 1. Here it is not explicitly stated, but one would assume day 1 for each patient to be the date of diagnosis of COVID-19. Many of the items listed as exclusion criteria would not have been knowable on that day. The authors may want to reconsider their exclusion criteria.

The authors should add how the diagnosis of COVID-19 was made.

For the purpose of this study, grouping into "mild" and "severe" disease appears to be based on whether or not patients required supplemental oxygen or ICU admission. This could be stated simply. It may be unnecessarily complicated to describe the MHLW guidelines when stating how severity was defined.

Discussion, line 201-202: It would be more correct to state that the study found an association between prior booster vaccination and lower severity of XBB-variant COVID-19 among hospitalized patients with COVID-19. This study design does not allow one to claim that the study revealed that bivalent omicron booster vaccination reduced the severity of COVID-19.

6. PLOS authors have the option to publish the peer review history of their article (what does this mean?). If published, this will include your full peer review and any attached files.

Reviewer #1: No

Reviewer #2: **Yes: **Nabin K. Shrestha

---

## [Author Response · Author response to Decision Letter 0]

2 Aug 2024

Thank you for reviewing our paper and for this suggestion. We performed a whole- genome analysis of stored nasopharyngeal specimens from patients admitted to our hospital and have added the distribution of sub-lineages to the manuscript. We have also changed the title. We have revised our manuscript to the best of our ability according to your suggestions. Please find our responses to your comments below. 

[C1] Lines 64: This phrase cannot clearly understandable “….XBB/XBB1.5 had more prior infections than…. “

(R1) Thank you for pointing this out. We have corrected the sentence (lines 68-71).

[C2] Fig 1. Should be explained in the result, but not in the introduction. The methods to collect the data need to be describe clearly in the methods. Otherwise, cite from the original publication.

(R2) Thank you for pointing this out. We have removed Fig 1 and have added the information as a reference because it is not part of the results of this study but was rather added to the Introduction as a description to help readers gain a better understand of the background of this study (line 82).

[C3] Line 97: There is some disagreement that antigen testing can be use as confirmatory diagnosis for COVID-19. Authors need to address this in discussion with solid references.

(R3) Thank you for pointing this out. To address this, we have stated as part of the limitations in the Discussion section that antigen testing is used at many facilities in Japan for the diagnosis of COVID-19 and is known to be as specific as PCR for omicron variants (lines 294-297).

[C4] Lines 99-100: Please describe the diagnosis criteria for bacterial pneumoniae. Is it supported with culture from blood or respiratory specimen? Or other methods such as NAAT?

(R4) As you have indicated, the diagnosis of bacterial pneumonia is supported by the results of sputum culture. The diagnostic criteria of bacterial pneumonia have been described in lines 101-103.

[C5] Line 46: How authors defined hospital acquired infection for COVID-19? Does the author want to explain that 38% of the subjects contracted with COVID-19 while they were stayed in the hospital because of suffering from other diseases? Or, it means that 38% of the patients have other HAIs (not the COVID-19)

(R5) Thank you for pointing this out. We defined hospital-acquired infection as new-onset COVID-19 during hospitalization for other diseases. We have described the criteria for hospital-acquired COVID-19 in lines 128-129.

[C6] Table 1. Why only male is presented?

(R6) Thank you for pointing this out. We added the data for female patients in Table 1. 

[C7] This study only recruited elderly (64 – 91) patients. Though it may reflect the actual hospitalization cases in Okinawa, the author should address these issues. Authors should refer to other countries reports which may have different segment of patients ages. This is important to give comprehensive insight into the characteristics, symptoms and sign, as well as risk factors for the XBB variant.

(R7) As you have indicated, most of the eligible patients in this study were older adults. We have added information regarding the differences in age distribution from that in other countries and need for further research on the effects on young adults to the Discussion (lines 232-237).

Reviewer #2: The study examined clinical features of COVID-19 caused by the XBB variant of SARS-CoV-2 and explored risk factors for severe disease among hospitalized patients.

Thank you for reviewing our paper. We have revised our manuscript to the best of our ability according to your suggestions. Please find our responses below.

[C1] The study was done in 4 tertiary care hospitals in Japan. Patients with complications related to bacterial infections were excluded.

Exclusion criteria appear to be complications that one would expect to occur in severe disease, so it is not clear why those who had these outcomes were excluded. In a cohort study, exclusions are made based on factors knowable before day 1 for each patient, not on factors that occur after day 1. Here it is not explicitly stated, but one would assume day 1 for each patient to be the date of diagnosis of COVID-19. Many of the items listed as exclusion criteria would not have been knowable on that day. The authors may want to reconsider their exclusion criteria.

(R1) Thank you for pointing this out. However, COVID-19 is often associated with bacterial co-infection or secondary bacterial infections, which are difficult to clearly identify. We considered that if symptoms and severity at diagnosis were obviously related to bacterial infection, it would greatly affect the evaluation of risk factors for severe COVID-19, which was the purpose of this study; therefore, we consulted with a statistics specialist before starting the study and added cases of clear bacterial infection to the exclusion criteria. We have added this intent to the limitations (lines 310-312). 

[C2] The authors should add how the diagnosis of COVID-19 was made.

(R2) Thank you for pointing this out. Accordingly, we have added a clear description of the diagnostic criteria for COVID-19 to the revised manuscript (lines 96-100).

[C3] For the purpose of this study, grouping into "mild" and "severe" disease appears to be based on whether or not patients required supplemental oxygen or ICU admission. This could be stated simply. It may be unnecessarily complicated to describe the MHLW guidelines when stating how severity was defined.

(R3) Thank you for your kind suggestion. We have corrected the Methods in the abstract and main body accordingly (lines 40-41, lines 131-133).

[C4] Discussion, line 201-202: It would be more correct to state that the study found an association between prior booster vaccination and lower severity of XBB-variant COVID-19 among hospitalized patients with COVID-19. This study design does not allow one to claim that the study revealed that bivalent omicron booster vaccination reduced the severity of COVID-19.

(R4) Thank you for your kind suggestion. We have corrected the sentence in the abstract and main body accordingly (lines 49-50, lines 220-221).

---

## [Decision Letter · Decision Letter 1]

20 Aug 2024

Clinical features of and severity risk factors for COVID-19 in adults during       

the predominance of SARS-CoV-2 XBB variants in Okinawa, Japan

PONE-D-24-16162R1

Dear Dr. Miyagi,

We’re pleased to inform you that your manuscript has been judged scientifically suitable for publication and will be formally accepted for publication once it meets all outstanding technical requirements.

Kind regards,

Liling Chaw

Academic Editor

PLOS ONE

Additional Editor Comments (optional):

Reviewers' comments:

Reviewer's Responses to Questions

**Comments to the Author**

1. If the authors have adequately addressed your comments raised in a previous round of review and you feel that this manuscript is now acceptable for publication, you may indicate that here to bypass the “Comments to the Author” section, enter your conflict of interest statement in the “Confidential to Editor” section, and submit your "Accept" recommendation.

Reviewer #1: All comments have been addressed

Reviewer #2: All comments have been addressed

2. Is the manuscript technically sound, and do the data support the conclusions?

Reviewer #1: Yes

Reviewer #2: Yes

3. Has the statistical analysis been performed appropriately and rigorously? 

Reviewer #1: I Don't Know

Reviewer #2: Yes

4. Have the authors made all data underlying the findings in their manuscript fully available?

Reviewer #1: Yes

Reviewer #2: No

5. Is the manuscript presented in an intelligible fashion and written in standard English?

Reviewer #1: Yes

Reviewer #2: Yes

6. Review Comments to the Author

Reviewer #1: (No Response)

Reviewer #2: (No Response)

7. PLOS authors have the option to publish the peer review history of their article (what does this mean?). If published, this will include your full peer review and any attached files.

Reviewer #1: No

Reviewer #2: **Yes: **Nabin K. Shrestha

---

## [Editor Report · Acceptance letter]

22 Oct 2024

PONE-D-24-16162R1 

PLOS ONE

Dear Dr. Miyagi, 

I'm pleased to inform you that your manuscript has been deemed suitable for publication in PLOS ONE. Congratulations! Your manuscript is now being handed over to our production team.

Kind regards, 

on behalf of

Dr. Liling Chaw 

Academic Editor

PLOS ONE